# Identification of oleoylethanolamide as an endogenous ligand for HIF-3α

Xiaotong Diao [1,8], Fei Ye [2,3,8], Meina Zhang[1], Xintong Ren[1], Xiaoxu Tian[4], Jingping Lu[5], Xiangnan Sun[1], Zeng Hou[6,7], Xiaoyu Chen[1], Fengwei Li[1], Jingjing Zhuang[1], Hong Ding[7], Chao Peng [4], Fraydoon Rastinejad[5✉], Cheng Luo [2,6,7✉] & Dalei Wu [1✉]

Hypoxia-inducible factors (HIFs) are α/β heterodimeric transcription factors modulating cellular responses to the low oxygen condition. Among three HIF-α isoforms, HIF-3α is the least studied to date. Here we show that oleoylethanolamide (OEA), a physiological lipid known to regulate food intake and metabolism, binds selectively to HIF-3α. Through crystallographic analysis of HIF-3 α/β heterodimer in both apo and OEA-bound forms, hydrogen-deuterium exchange mass spectrometry (HDX-MS), molecular dynamics (MD) simulations, and biochemical and cell-based assays, we unveil the molecular mechanism of OEA entry and binding to the PAS-B pocket of HIF-3α, and show that it leads to enhanced heterodimer stability and functional modulation of HIF-3. The identification of HIF-3α as a selective lipid sensor is consistent with recent human genetic findings linking HIF-3α with obesity, and demonstrates that endogenous metabolites can directly interact with HIF-α proteins to modulate their activities, potentially as a regulatory mechanism supplementary to the well-known oxygen-dependent HIF-α hydroxylation.

[1] Helmholtz International Lab, State Key Laboratory of Microbial Technology, Shandong University, 266237 Qingdao, China. [2] School of Pharmaceutical Sciences, Zhejiang Chinese Medical University, 310053 Hangzhou, China. [3] College of Life Sciences and Medicine, Zhejiang Sci-Tech University, 310018 Hangzhou, China. [4] National Facility for Protein Science in Shanghai, Zhangjiang Lab, Shanghai Advanced Research Institute, Chinese Academy of Science, 201210 Shanghai, China. [5] Target Discovery Institute, NDM Research Building, University of Oxford, Old Road Campus, Oxford OX3 7FZ, UK. [6] School of Pharmaceutical Science and Technology, Hangzhou Institute for Advanced Study, University of Chinese Academy of Sciences, 310053 Hangzhou, China. [7] Drug Discovery and Design Center, the Center for Chemical Biology, State Key Laboratory of Drug Research, Shanghai Institute of Materia Medica, Chinese Academy of Sciences, 201203 Shanghai, China. [8] These authors contributed equally: Xiaotong Diao, Fei Ye. ✉email: fraydoon.rastinejad@ndm.ox.ac.uk; cluo@simm.ac.cn; dlwu@sdu.edu.cn

Hypoxia-inducible factors (HIFs) belong to the basic helix-loop-helix PER-ARNT-SIM (bHLH-PAS) family of transcription factors, which sense physiological changes or environmental stimuli and respond by modulating target gene expression[1]. Like other members of this family, transcriptionally active HIFs are heterodimeric proteins consisting of one α subunit and one β subunit[2]. There are three HIF-α isoforms in mammals (HIF-1α, HIF-2α, and HIF-3α), each encoded by a different gene. The β subunit, also known as aryl hydrocarbon receptor nuclear translocator (ARNT), is a Class II member within the bHLH-PAS family, and it acts as a common heterodimerization partner for certain Class I members that include the HIF-α proteins, aryl hydrocarbon receptor, neuronal PAS (NPAS) proteins and single-minded (SIM) proteins[2].

The biological functions of HIF-1α and HIF-2α have been investigated extensively[3,4], since the initial cloning and early characterization of these two genes in 1995 and 1997, respectively[5–9]. The oxygen-dependent hydroxylation of specific proline residues on HIF-α by hydroxylases, leads to its proteasomal degradation under normoxia[10–12]. In addition, an asparagine at the C-terminus of HIF-α can be hydroxylated under normoxia[13], which blocks its interaction with coactivators necessary for the transcriptional activation[14]. These two oxygen-dependent protein modifications both restrict HIF-α activities at normal oxygen levels. But under oxygen deprivation, HIF-α proteins escape degradation, dimerize with ARNT and initiate the transcription of gene programs leading to increased angiogenesis, erythropoiesis and glycolytic metabolism, in an isoform- and cell type-specific manner[3]. Besides oxygen level, the HIF pathway is also regulated by other cellular conditions, such as the metabolic status[15]. For example, the tricarboxylic acid (TCA) cycle intermediate α-ketoglutarate is an obligate substrate required for the hydroxylase-dependent HIF-α modification[4]. However, the clear-cut cavities inside the PAS domains of HIF-α and ARNT proteins raise the key question of whether endogenous small-molecule ligands, such as metabolites, can directly bind to and modulate the activity of HIFs[16,17].

Compared with HIF-1α and HIF-2α, HIF-3α is a much less studied isoform[18], despite having its gene first identified in 1998. A complication to understanding the full range of HIF-3α functions is that multiple variants of different sizes appear to be produced from its gene locus, due to alternative mRNA splicing and utilization of

different promoters or transcription initiation sites[19]. At least seven such HIF-3α variants were confirmed in human cells (Supplementary Fig. 1), and three variants found in mouse[20]. Among these variants, two types have been investigated most. The first type includes the longer HIF-3α variants that share a similar domain composition with full-length HIF-1α and HIF-2α proteins, except that a C-terminal leucine zipper (LZIP) domain replaces the C-terminal transactivation domain[21]. With this LZIP substitution, these HIF-3α variants can dimerize with ARNT properly but exhibit a relatively lower transcriptional activity[22]. Recent studies have revealed that certain long variants of human HIF-3α function as transcription activators and upregulate their own specific target genes. For example, overexpression of HIF-3α2 in Hep3B cells induces the expression of a group of genes[23], including erythropoietin (EPO) and heat shock protein family A (Hsp70) member 6 (HSPA6). In addition, overexpression of HIF-3α9 in HEK293 cells upregulates the expression of REDD1, LC3C, and SQRDL[24]. The second type of HIF-3α variants are quite distinct. Through alternative splicing, their polypeptide chains are only about half the size (e.g., HIF-3α4 in Supplementary Fig. 1), with a special sequence in the PAS-B domain that has been suggested to impart a unique ability to inhibit the HIF pathway activity through direct binding to HIF-1α and HIF-2α proteins[25,26].

Interestingly, the potential role of HIF-3α as a lipid sensor was suggested mainly based on the crystal structure of its PAS-B domain in complex with bacterial lipids, which remained bound within the cavity during protein expression, purification and crystallization[27]. Here, we began by examining the overall heterodimeric structure of HIF-3α-ARNT complex, consisting of the N-terminal segments (including the bHLH, PAS-A and PAS-B domains as shown in Fig. 1a) from both proteins. We then conducted a biochemical high-throughput screening campaign using a compound library of cellular metabolites, leading to the identification of an endogenous lipid oleoylethanolamide (OEA) as a ligand capable of binding to HIF-3α selectively. OEA is a naturally occurring ethanolamide produced in the small intestine following feeding, and regulates satiety and body weight[28]. We further studied the molecular mechanisms of how OEA enters the closed PAS-B pocket of HIF-3α, and how that binding enhances HIF-3α-ARNT heterodimer stability and transcriptional activity in cells.

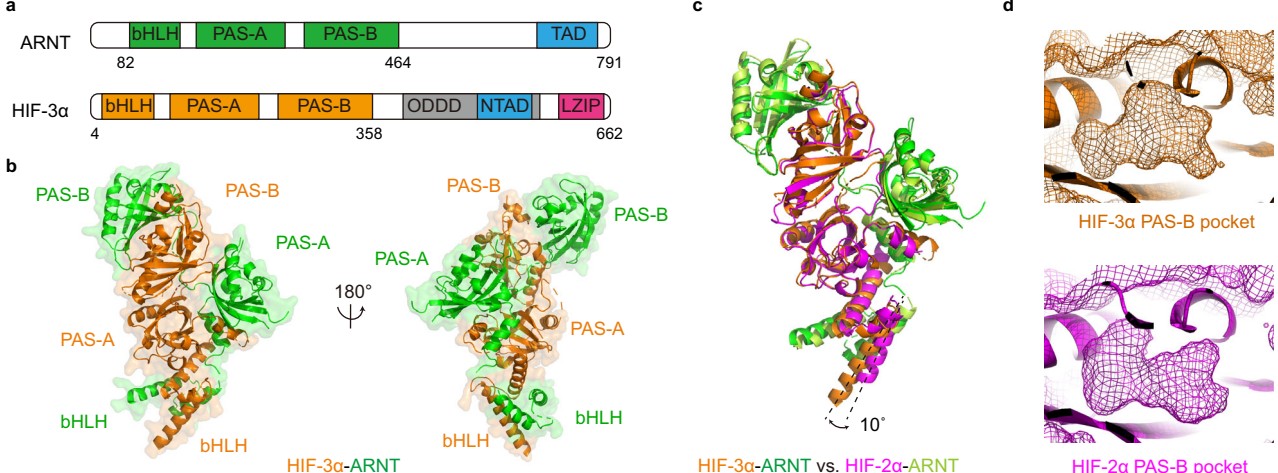

**Fig. 1 Crystal structure of the HIF-3α-ARNT heterodimer in *apo* form. a** Schematic representation showing the domain arrangements of ARNT and HIF-3α. bHLH basic helix-loop-helix, PAS PER-ARNT-SIM, ODDD oxygen-dependent degradation domain, TAD transactivation domain, LZIP leucine zipper. **b** Overall structure of the HIF-3α-ARNT heterodimer in two views with all domains labelled. HIF-3α and ARNT are colored in orange and green, respectively. **c** Structure superimposition of HIF-3α-ARNT and HIF-2α-ARNT. **d** The shape of empty PAS-B cavities of HIF-3α (orange) and HIF-2α (magenta) shown in mesh.

## Results

**Quaternary structure of HIF-3α-ARNT heterodimer.** To fully visualize the ligand-binding potential of HIF-3α, we crystallized the multi-domain mouse HIF-3α-ARNT complex and solved its structure at 2.3 Å resolution (Fig. 1b). Each asymmetric unit of the crystal contained a single heterodimer (Supplementary Table 1). The overall structure of HIF-3α-ARNT proved to be fairly similar to that of the HIF-2α-ARNT complex[17], with a root mean squared deviation (RMSD) of 1.8 Å between their Cα atoms. Meanwhile as shown in the superimposition of these two structures (Fig. 1c), the N-terminal α helix in the bHLH domain of HIF-3α is slightly longer and rotated about 10° relative to that of HIF-2α. However, it remains unclear if this distinction is real or artificial, as it may have risen only due to the differences in their respective crystal packings that impact this region.

In the previously reported high-resolution structure of the single human HIF-3α PAS-B domain[27], its inner cavity was occupied by a mixture of lipids that derived from the *Escherichia coli* expression system. In our study, protein production of the multi-domain HIF-3α-ARNT heterodimer in *E. coli* only yielded an apo form with no discernible ligand present in the PAS-B domain of HIF-3α or elsewhere in the complex. The contrasting finding about the pocket contents of overexpressed HIF-3α proteins suggests that although bacteria-derived lipids can bind to HIF-3α, their ease of removal during purification is consistent with their reversible binding capability. As calculated by Fpocket program[29], we didn't find a sizeable pocket in the HIF-3α PAS-A domain, but observed an empty cavity of about 280 Å³ within the PAS-B domain, comparable in size to the corresponding HIF-2α pocket (Fig. 1d). This vacancy provided us with a clear opportunity to screen small-molecule libraries to identify unique molecules capable of binding to HIF-3α selectively.

**Identification of OEA as a direct-binding ligand for HIF-3α.** To discover novel ligands interacting with HIF-3α, we conducted a biochemical direct-binding assay using the affinity selection-mass spectrometry (AS-MS) technique[30] with purified multi-domain HIF-3α-ARNT heterodimer proteins. Aiming at potential intrinsic HIF-3α ligands, we screened an in-house collection of more than 2000 cellular endogenous molecules, with >7% representing fatty acids and lipids. The screen identified OEA, a natural metabolite of oleic acid to be a highly selective binder of HIF-3α (Fig. 2a). OEA has been found in different tissues, but its mobilization due to dietary fat intake is known to occur in the intestine after feeding[31,32]. OEA supplementation has been shown to regulate food intake and consumption in human subjects and in rats[33]. The biological functions of OEA in terms of controlling lipid and glucose metabolism have also been described[34].

In our chemical screens, we did not detect OEA binding to either HIF-1α-ARNT or HIF-2α-ARNT heterodimers, which both served as counter-screening targets, indicating the binding of OEA to HIF-3α-ARNT to be selective and further suggesting that this binding was directed to the HIF-3α subunit (since all three heterodimers contained an identical ARNT protein). No other selective ligands whose binding could be reconfirmed using an orthogonal assay were identified in our screening, for any of the three HIF-α subunits. To confirm that OEA indeed binds to HIF-3α directly, we first conducted a thermal shift assay with the HIF-1α-ARNT, HIF-2α-ARNT and HIF-3α-ARNT complex proteins, respectively (Fig. 2b). OEA increased the melting temperature ($T_m$) of the HIF-3α-ARNT complex by about 1.7 °C, much higher than its effects on the other two isoforms. Subsequently, we employed the surface plasmon resonance (SPR) method to confirm and measure the binding affinity of OEA to the isolated HIF-3α PAS-B domain, and obtained an approximate

$K_D$ value of 14.0 μM (Fig. 2c). These findings pinpointed the PAS-B domain pocket of HIF-3α as the binding site of OEA.

HIF-2α antagonists targeting its PAS-B domain have been found to allosterically disrupt the interactions between HIF-2α and ARNT[35–37]. We previously discovered HIF-2α agonists and revealed the molecular mechanism of how ligands binding to HIF-2α's PAS-B pocket bidirectionally modulate its dimerization with ARNT[37]. To test for this possibility in the context of the HIF-3α-ARNT complex, we used a time-resolved fluorescence resonance energy transfer (TR-FRET)-based in vitro assay that we established and utilized for heterodimeric HIF proteins[37]. As shown in Fig. 2d, OEA enhanced the interaction between HIF-3α and ARNT, while it had no discernible effect on the interaction of HIF-2α with ARNT. These results suggest that OEA specifically binds to HIF-3α and promotes the physical dimerization between HIF-3α and ARNT proteins, and thus potentially functions as a HIF-3α agonist.

We then sought to test if there is an agonistic effect of OEA on the transcriptional activity of HIF-3α. As mentioned earlier, the human *HIF-3A* gene undergoes extensive alternative splicing, resulting in a number of variants with different lengths or sequences (namely HIF-3α1 ~ HIF-3α10, Supplementary Fig. 1). In this study we focused on the representative variant HIF-3α1, which possesses the bHLH, PAS-A, PAS-B, ODDD (oxygen-dependent degradation domain), NTAD (N-terminal transactivation domain) and LZIP regions in its polypeptide chain. First, we overexpressed HIF-3α1 in HEK293 cells and checked the mRNA expression level of several reported target genes[23,24], including *EPO*, *HSPA6*, *REDD1*, *LC3C* and *SQRDL*. Among them, the expression of *HSPA6* was clearly elevated by HIF-3α1 over-expression in the HEK293 cells (Fig. 2e), as well as in the Hep3B and HepG2 cells (Supplementary Fig. 2a, b), suggesting that *HSPA6* might be a downstream gene positively regulated by the HIF-3α1 variant. Next, we treated the HEK293 cells with OEA, and found that OEA further increased the *HSPA6* level with HIF-3α1 overexpression (Fig. 2e), indicating that OEA may function as an agonist to enhance the activity of HIF-3α. To explore for more HIF-3α target genes, we conducted a preliminary RNA-seq using HEK293 cells and confirmed the mRNA expression of several genes by real-time PCR. At least three genes, *TCF20*, *PSME2* and *IFIT1*, were identified as potential downstream genes of the HIF-3α pathway, as their mRNA levels were elevated by HIF-3α1 overexpression and further by OEA treatment (Supplementary Fig. 2c).

Since the above-mentioned cell-based experiments were performed under a normal oxygen condition, we wanted to know whether or to what extent the overexpressed HIF-3α1 proteins would be degraded in cells. Therefore, we tested the HIF-3α protein level within HEK293 cells that were transfected with an empty vector, the full-length wide-type HIF-3α1 and its corresponding degradation-resistant P490A mutant[38], respectively. According to the Western blotting results (Supplementary Fig. 2d), the intrinsic HIF-3α protein was barely detectable under normoxia in these cells. Surprisingly, the overexpressed wild-type HIF-3α1 reached a similar level as the P490A mutant, indicating that the oxygen-dependent degradation was not decisive for HIF-3α in our experimental conditions. Moreover, OEA treatment showed no discriminable effect on the HIF-3α protein level in cells, despite its facilitation on the dimerization between HIF-3α and ARNT biochemically (Fig. 2d).

**Agonistic mechanism of OEA through allosteric effects.** To further explore the mechanism of binding and HIF-3α transcriptional modulation by OEA, we pursued and obtained the crystal structure of HIF-3α-ARNT in complex with OEA at a

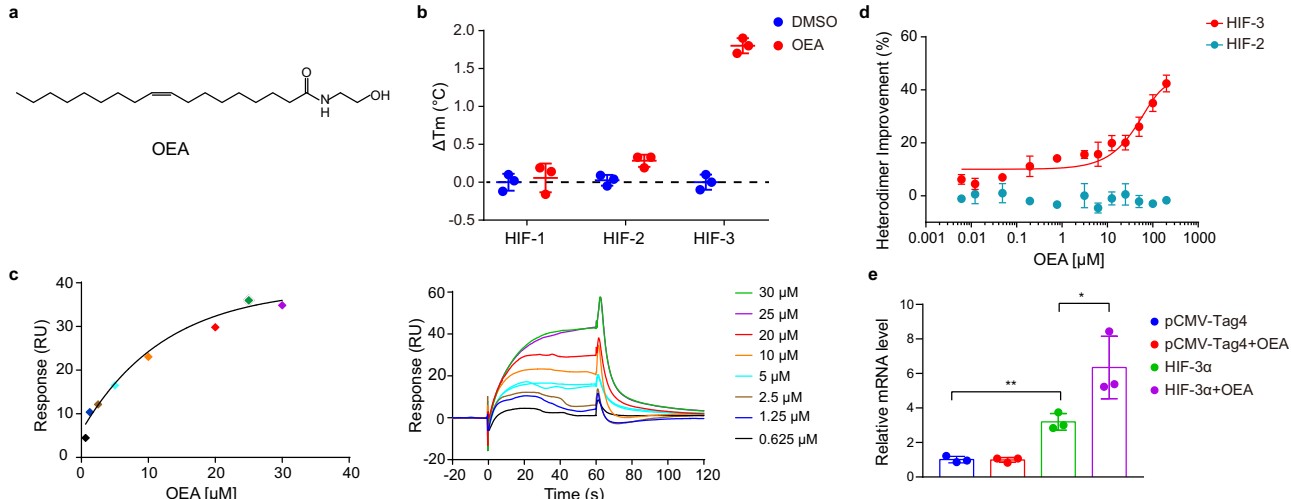

**Fig. 2 Agonistic effects of OEA on the HIF-3α-ARNT heterodimer. a** Chemical structure of OEA. **b** Protein thermal shift assays measuring the change in melting temperatures (Δ$T_m$) of HIF-1α-ARNT, HIF-2α-ARNT and HIF-3α-ARNT (i.e., HIF-1, HIF-2 and HIF-3) complex proteins in the presence of 20 μM OEA. **c** Binding of OEA to the HIF-3α PAS-B domain ($K_D$ about 14.0 μM) as measured by SPR. **d** Effects of OEA on the heterodimerization of HIF-3α-ARNT and HIF-2α-ARNT, detected by TR-FRET. **e** The effects of HIF-3α overexpression and 25 μM OEA treatment on the expression of *HSPA6* in HEK293 cells. **p = 0.0019 for pCMV-Tag4 vs. HIF-3α, *p = 0.0437 for HIF-3α vs. HIF-3α + OEA. Error bars, mean ± SD.; n = 3 (biological replicates) for (**d,e**). Statistical significance: *p < 0.05, **p < 0.01 (Unpaired two-tailed t test).

2.5 Å resolution (Fig. 3a, Supplementary Table 1). This complex revealed the stereochemical basis for the direct binding of OEA to HIF-3α PAS-B cavity. Its fatty acid chain deeply inserted into the pocket and the ethanolamine partially exposed outside the pocket (Fig. 3b), as clearly seen in the "omit" electron density maps (Supplementary Fig. 3a). Mainly through hydrophobic interactions, OEA contacts with multiple residues in the HIF-3α PAS-B pocket, including F239, H243, M247, F249, I276, V284, I288, L291, Y302, F304, and A318 (Fig. 3b). In addition, OEA forms two hydrogen bonds with the main chain atoms of H336 and L338 near the pocket entrance (Fig. 3b).

To learn how OEA binding changes the conformation of HIF-3α-ARNT heterodimer, we compared the OEA-bound and apo structures directly (Supplementary Fig. 3b). By superimposition, their overall dimeric structures are very similar with a RMSD of only 0.4 Å for all Cα atoms. However, there are two regions showing clear conformational changes. One locates at the PAS-B pocket of HIF-3α, where residues G237, F239 and L338 move their positions substantially to accommodate OEA binding (Fig. 3c, Supplementary Fig. 3c). As a result, the size of PAS-B pocket dramatically increased after the binding of OEA, from about 280 Å³ in the apo form, to more than 700 Å³ in the complex bound to OEA, suggesting that OEA binding increases the cavity size by 150% (Supplementary Fig. 3d, e). We noticed that the cavity size within the single PAS-B domain structure of HIF-3α was reported as only 510 Å³ even in the presence of bound lipid[27]. For a direct comparison, we calculated the PAS-B pocket of their structure using Fpocket[29] and obtained a similar value of 700 Å³. Since the cavity size measurement would depend on the algorithms of different programs, we adopted another program called PyVOL[39] for a recalculation. The HIF-3α PAS-B pocket of our structure in the apo and OEA-bound forms was measured respectively as 225 Å³ and 516 Å³, which again showed that this cavity expands substantially upon ligand binding. The second region effected is the loop connecting ARNT PAS-A and PAS-B domains (A/B loop), which is more stable and continuous in the OEA-bound complex structure (Fig. 3d), as evidenced by the differences in their "omit" density maps (Supplementary Fig. 3f, g). This stabilization effect of OEA on the A/B loop is consistent with its ability to stabilize the HIF-3α-ARNT heterodimer (Fig. 2d).

To further compare the intrinsic dynamics of HIF-3α-ARNT in the OEA-bound and apo forms, we conducted MD simulations on three systems (Supplementary Fig. 4a–c): HIF-3α-ARNT in complex with OEA (HIF-3$^{OEA}$), HIF-3α-ARNT-OEA complex excluding OEA (HIF-3$^{noOEA}$) and apo HIF-3α-ARNT structure (HIF-3$^{apo}$). The root mean square fluctuation (RMSF) profile of ARNT subunit suggests that besides the modelled segments, the ARNT A/B loop shows a high degree of flexibility, especially in the context of the HIF-3$^{noOEA}$ system (Fig. 3e). We further monitored the RMSD values for ARNT A/B loop in three systems (Fig. 3f), which revealed that A/B loop in HIF-3$^{OEA}$ is more stable than that in the other two systems, validating this region is stabilized by OEA binding.

We also found that majority of HIF-3α residues within the PAS-B pocket undergo only subtle conformational changes upon OEA binding, except for the three residues G237, F239 and L338 that gate the inner cavity and enlarge the pocket by conformational changes (Fig. 3c, Supplementary Fig. 3c–e). However, it is difficult to explain the role of OEA in promoting the stability of HIF-3α-ARNT heterodimer, solely based on the movement of these three residues. To fully investigate the contributions of individual residues locating at the interface between A/B loop and HIF-3α PAS-B domain to the dimerization energy, we carried out a per-residue decomposition study of relative binding energy based on MD simulations using the MM-GBSA method (excluding entropic contribution, Supplementary Table 2). As shown in Fig. 3g, in OEA-bound HIF-3α-ARNT, the middle part of ARNT A/B loop, the N-terminal region of Fα, and the Gβ from HIF-3α PAS-B together contribute to the stabilization of the dimer, especially for the hydrogen bonds between ARNT A/B loop and HIF-3α residue R303, which disappear in last 150 ns in the MD trajectory of HIF-3$^{noOEA}$ (Fig. 3h, Supplementary Fig. 4d). These data suggest that the interface between ARNT A/B loop and the Fα and Gβ regions of HIF-3α, is very critical to mediate the allosteric effects of OEA on dimerization.

Next, to experimentally monitor conformational changes of HIF-3α-ARNT in the solution state, we employed the HDX-MS assay and examined the altered protein dynamics in the presence of OEA. HDX-MS data indicated that OEA binding stabilized multiple residues within the Gβ and Fα regions of HIF-3α PAS-B domain (Fig. 3i, Supplementary Fig. 5a), consistent with results

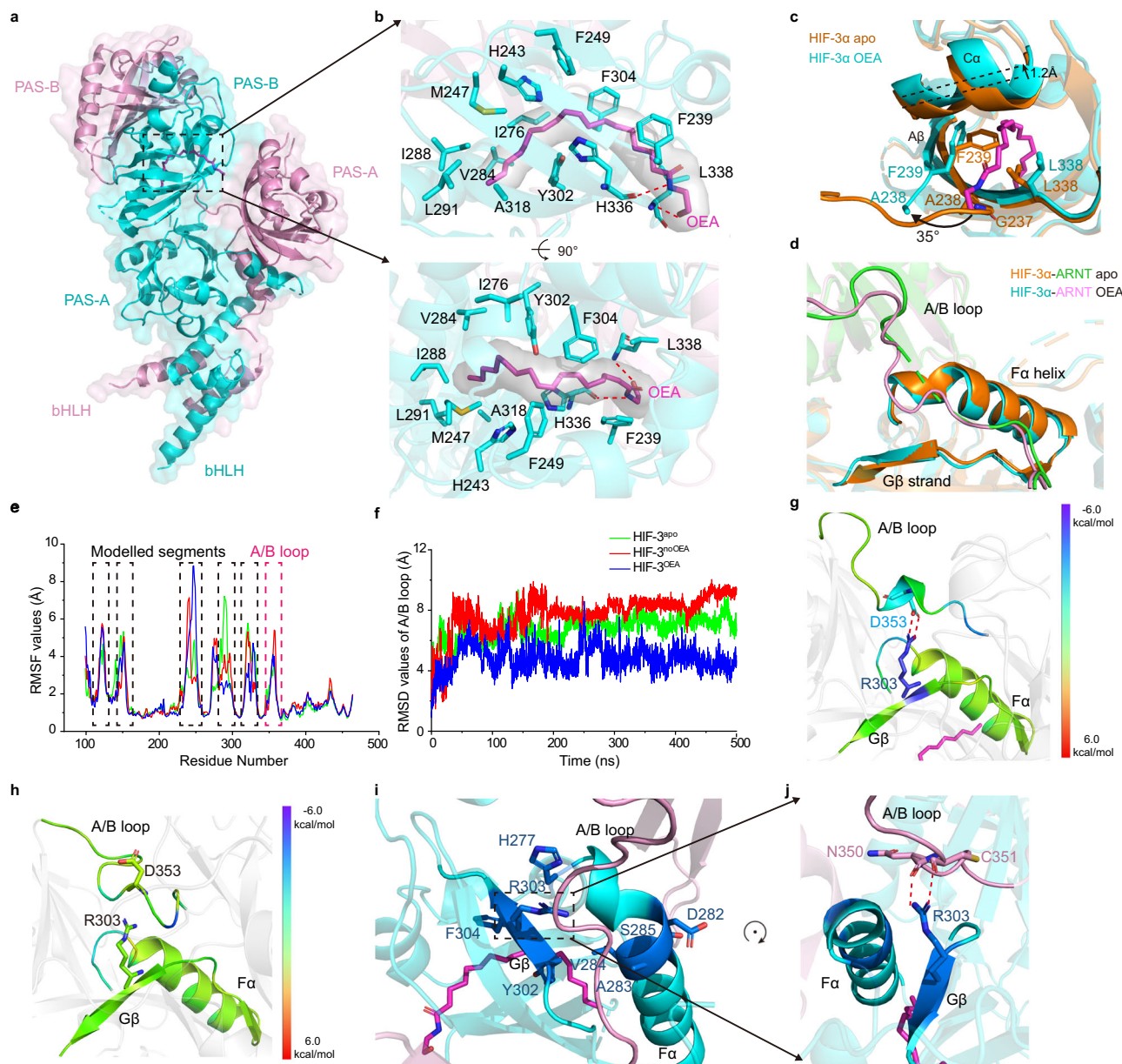

**Fig. 3 The molecular mechanism of OEA on stabilizing the HIF-3α-ARNT heterodimer. a** Overall crystal structure of the HIF-3α-ARNT in complex with bound OEA. HIF-3α and ARNT are colored in cyan and pink, respectively. **b** An enlarged view of OEA (magenta) inside its binding pocket, with residues involved in the interaction shown as sticks. **c,d** Superimposition of HIF-3α-ARNT structures with and without OEA, at the entrance position of HIF-3α PAS-B domain (**c**) and the ARNT A/B loop (**d**), highlighting the conformational changes after OEA binding. HIF-3α and ARNT proteins in the apo form are colored in orange and green, respectively. **e** Cα RMSF plots for the ARNT subunit in each system. The highly flexible domains including modelled linkers (black) and A/B loop (magenta) are highlighted. **f** Cα RMSD plots for the ARNT A/B loop (residues 349-360) in each system. **g,h** Per-residue decomposition of relative binding energy for residues within the interface between ARNT A/B loop and HIF-3α PAS-B domain in HIF-3^OEA (**g**) and HIF-3^noOEA (**h**) systems, respectively. For each system, the 500-ns snapshot is shown. **i,j** HDX-MS results mapped on the Fα and Gβ regions of HIF-3α. The residues with significantly reduced deuteration levels upon OEA binding are colored in dark blue (−90 to −50%) as compared to the apo form (**i**) (detailed in Supplementary Fig. 5a), and the R303 in Gβ of HIF-3α forms hydrogen bonds with A/B loop of ARNT (**j**). Hydrogen bonds are shown in red dotted lines.

from the above MD studies. To be more specific, OEA stabilized residues F304, Y302 and V284 by hydrophobic interactions, and thus immobilized Gβ and Fα. It is possible that the stabilized Gβ and Fα function as an "interlayer" to further contact with the AB-loop of ARNT. As shown in Fig. 3j, stabilized R303 at Gβ forms hydrogen bonds with ARNT A/B-loop. Therefore, results from HDX-MS, crystallographic and MD studies, together indicate that OEA allosterically enhances the interactions between HIF-3α and ARNT, by stabilizing key residues within the Gβ and Fα regions of HIF-3α PAS-B domain.

**Ligand entry route into the PAS-B pocket**. In the OEA-bound HIF-3α-ARNT co-crystal structure, we find that OEA was not completely shielded inside the HIF-3α PAS-B pocket. While its hydrophobic fatty acid chain was well buried in the pocket, the ethanolamine group of OEA was partially exposed to solvent outside of the cavity (Fig. 4a). This arrangement suggests a highly favorable binding mode whereby hydrophobic and hydrophilic portions are ideally disposed for enhanced overall binding energetics. An enlarged tunnel was formed within the HIF-3α PAS-B domain to accommodate the unique shape and size of the mono-

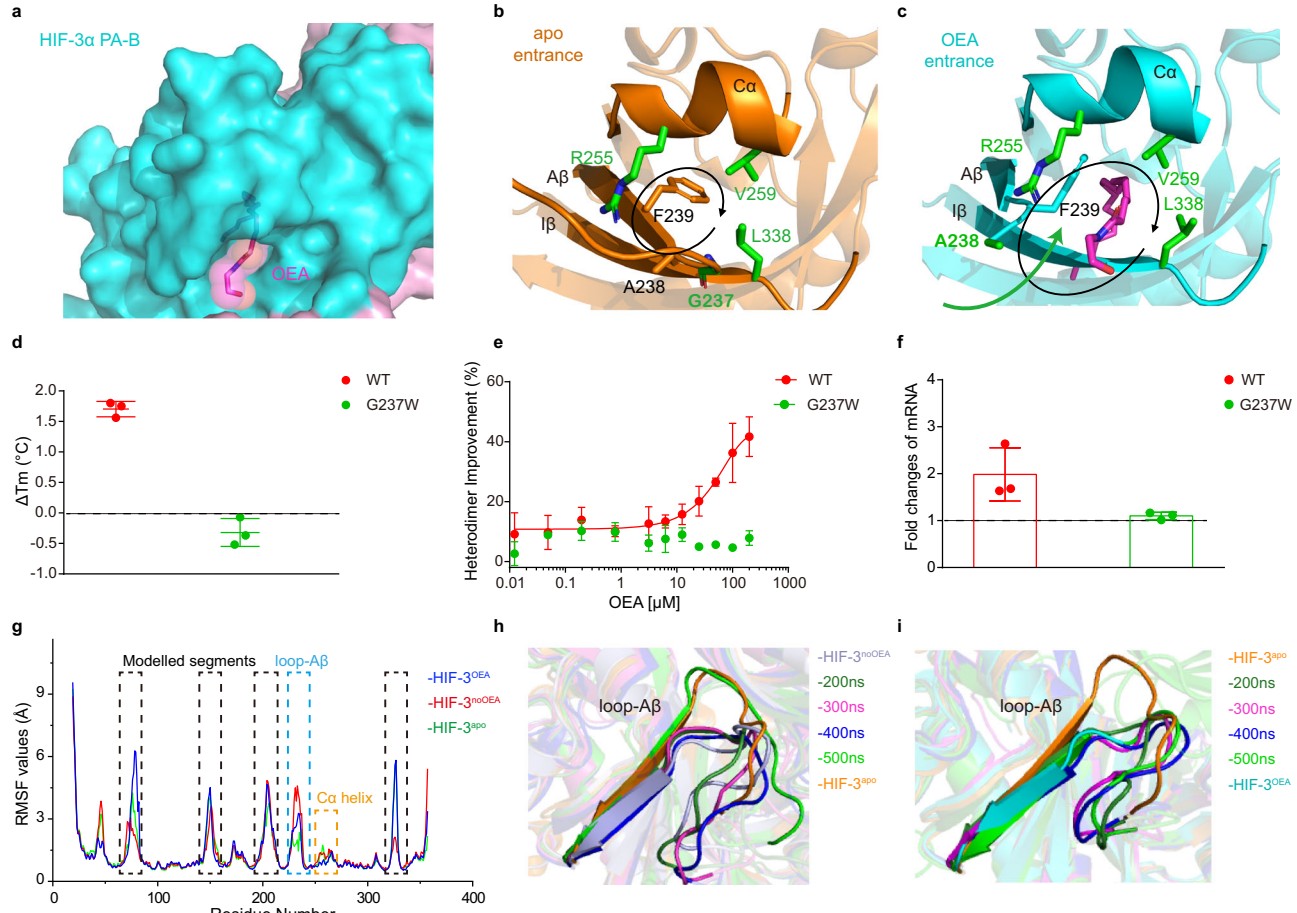

**Fig. 4 The loop-Aβ region of HIF-3α gates the OEA pocket entrance. a** An enlarged view of OEA in the binding pocket of HIF-3α, shown in the magenta. **b,c** Comparison of the HIF-3α PAS-B entrance position in the *apo* (**b**) and OEA-bound structures (**c**). The entry route of OEA is simply marked with a green arrow. The residues surrounding the entrance are labeled in green, and all residues are shown with their side chains, except G237. **d** The $\Delta T_m$ values of wild-type HIF-3α (WT) and G237W mutant both heterodimerized with ARNT were measured by PTS in the presence of 20 μM OEA. **e** The effect of OEA on the heterodimerization between HIF-3α mutant G237W and ARNT, as monitored by TR-FRET. **f** Comparison of fold changes in the mRNA expression of *HSPA6* between overexpressed wild-type HIF-3α and the mutant G237W after OEA (25 μM) treatment. **g** Cα RMSF plots for the HIF-3α subunit in each system. The highly flexible regions including modelled linkers (black) and loop-Aβ (cyan) are highlighted. And the region of Cα helix marks in orange. **h, i** Snapshot structures extracted from the trajectories of HIF-3$^{noOEA}$ (**h**) and HIF-3$^{apo}$ (**i**). Error bars, mean ± SD.; $n = 3$ (biological replicates) for (**d-f**).

unsaturated OEA molecule (Supplementary Fig. 3d, e), and this tunnel also suggests a clear entry route for OEA binding. As shown in Fig. 4b, the tunnel entrance was mainly surrounded and sealed by four residues, G237 at Aβ, R255 at Cα helix, V259 at Cα helix, and L338 at Iβ strand, in the apo HIF-3α structure. But upon OEA binding, the side chain of L338 flipped 90°, Cα helix moved 1.2 Å outward, and Aβ flipped by 35° to the outside of the pocket, thereby providing access to OEA within this entrance tunnel (Fig. 3c, Fig. 4c).

To verify the hypothesis that OEA enters the HIF-3α PAS-B pocket through the above route, we made a G237W mutation on HIF-3α, expected to block the entrance due to the bulky side chain of tryptophan. As implied by the decreased $\Delta T_m$ value in the thermal shift assay (Fig. 4d), the binding ability of OEA to this HIF-3α mutant was indeed reduced, compared with the wild-type protein. Moreover, the G237W mutation diminished the ability of OEA to enhance the stabilization of HIF-3α-ARNT heterodimer as monitored by the TR-FRET assay (Fig. 4e), as well as its ability to further induce the mRNA expression of *HSPA6* as measured by PCR (Fig. 4f). These results suggested that the loop region prior to Aβ, where G237 is located, is crucial for the entering and binding of OEA into the HIF-3α PAS-B pocket.

As shown in the comparison of superposed structures (Fig. 3c), the binding position of OEA overlaps with the Aβ of PAS-B domain from the apo HIF-3α structure. Therefore, the opening of Aβ is likely a key step for OEA to enter the pocket. RMSF values were calculated for the HIF-3α subunit in each of the three MD systems (Fig. 4g), indicating that besides modelled linkers and terminal positions, the Aβ with its prior loop (loop-Aβ, residues 227-244) and Cα helix regions, which surround the entrance of HIF-3α PAS-B pocket, exhibit changes in flexibility between different systems.

We then examined snapshots isolated from the trajectories of HIF-3$^{noOEA}$ and HIF-3$^{apo}$ respectively, showing that the region of loop-Aβ switches between two different states (Fig. 4h, i). The RMSD profile validated that this loop-Aβ region undergoes dramatic conformational changes between different states (Supplementary Fig. 6a). During 400–500 ns, in which this region exhibits the highest RMSD values, a small pocket appears at the entrance of OEA binding tunnel (Supplementary Fig. 6b). However, it disappears afterwards since this region switches back to the HIF-3$^{apo}$ state. In addition, we performed a 500 ns MD simulation on monomeric HIF-3α from the HIF-3$^{apo}$, which also revealed the flexibility of loop-Aβ region (Supplementary

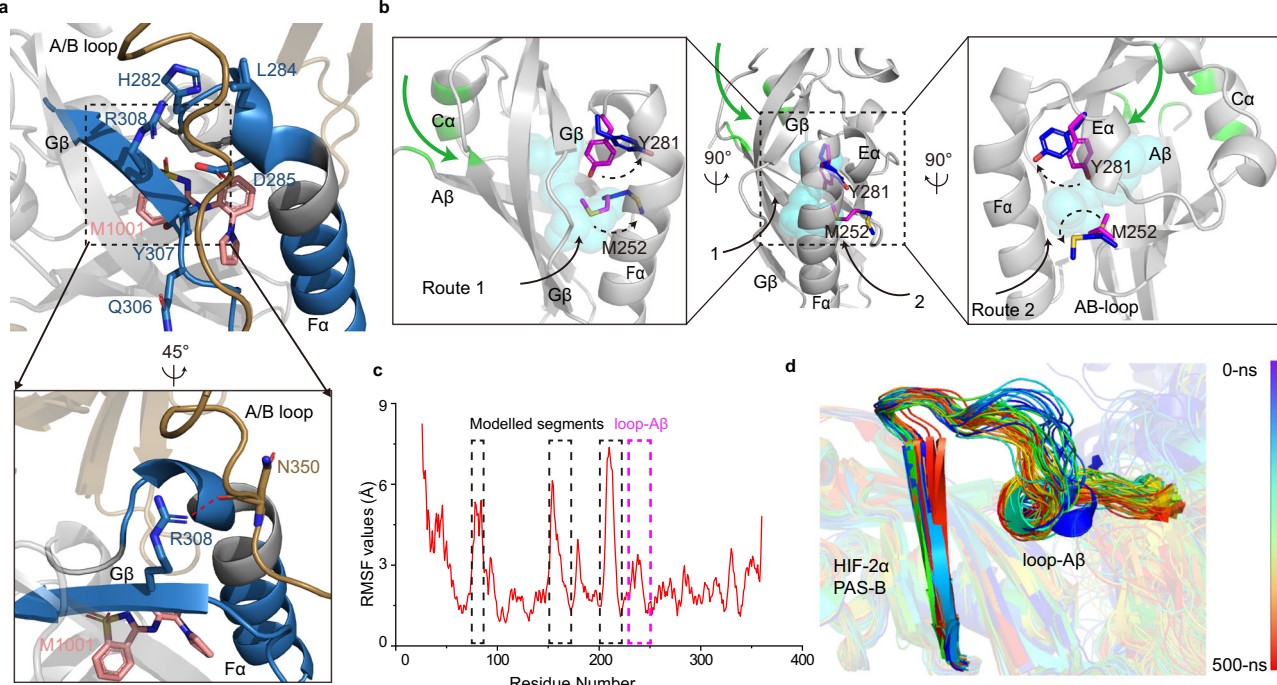

**Fig. 5 A possible common mechanism in ligand binding shared by HIF-2α. a** The residues of Fα and Gβ with reduced deuteration levels are mapped on the crystal structure in blue, and the R308 in Gβ of HIF-2α forms a hydrogen bond with the A/B loop of ARNT. Hydrogen bonds are shown in red dotted lines. **b** Two routes previously proposed for ligands to enter or leave the HIF-2α PAS-B pocket are indicated by black arrows, and enlarged views of two routes are shown on the left and right sides, respectively. The possible new entry route of ligands into HIF-2α, corresponding to that of OEA into HIF-3α, is shown by a green arrow, with the residues surrounding the entrance also colored green. M252 and Y281 are shown in magenta in the apo structure, and blue in co-crystal structures. M252 and Y281 were flipped outwards (as shown by dotted arrows) by antagonist PT2385 and agonist M1001, respectively. The PDB codes of HIF-2α-ARNT-PT2385 and HIF-2α-ARNT-M1001 are 6E3S and 6E3U. **c** Cα RMSF plots for HIF-2α in a monomeric system. The highly flexible regions including modelled linkers (black) and loop-Aβ (magenta) are highlighted. **d** The conformational change of the loop-Aβ region in snapshot structures extracted from the trajectories of HIF-2α during the MD stimulations.

Fig. 6c,d). These above MD results imply that the loop-Aβ region can sweep between different conformational states, leading to the generation of a small pocket at the entrance of HIF-3α PAS-B domain, which can then become subsequently enlarged and merged into the inner cavity during the entry of ligands.

**A common allosteric mechanism of HIF-α isoforms.** Since the protein sequences of three HIF-α isoforms are highly conserved (Supplementary Fig. 7), and their overall structural arrangements as heterodimers with ARNT are also similar[17] (Fig. 1c), it is possible that they all share a similar allosteric mechanism triggered by ligand binding to their PAS-B domains. We previously discovered novel agonists for HIF-2α, and identified the key residue Y281 that mediates the enhanced interaction at the dimer interface of HIF-2α and ARNT[37]. Interestingly, the HIF-2α agonist M1001 was also found to promote the stability of ARNT A/B loop in the structure of HIF-2α-ARNT complex[37]. And the HDX-MS data of this heterodimer revealed that M1001 enhanced the stability of Gβ and Fα in solution (Supplementary Fig. 5b). In addition, the HIF-2α residue R308 at Gβ, which is conserved in all three HIF-α proteins (corresponding to the HIF-3α R303), formed a hydrogen bond with the ARNT A/B loop in the complex structure (Fig. 5a, Supplementary Fig. 7). Therefore, we speculate that agonists binding to the HIF-α PAS-B pockets can allosterically enhance interactions between the Gβ and Fα regions of HIF-α and the A/B loop of ARNT, which might be a common mechanism for agonists to stabilize HIF-α-ARNT heterodimers.

Because both agonists and antagonists of HIF-2α were previously found to be totally encapsulated in the PAS-B domain in the co-crystal structures[37], it was unclear what the actual

ligand-entry pathway might be for the HIF-2α PAS-B pocket. A previous study using MD stimulations suggested two possible primary routes for ligands to enter or leave the HIF-2α PAS-B domain[40]. One is between Gβ and Fα (Route 1), and the other is between Fα, Eα and AB-loop (Route 2) (Fig. 5b). We previously identified two critical residues, Y281 and M252 within the PAS-B pocket of HIF-2α, whose side chains can be flipped out toward the PAS-B/PAS-B dimer interface by ligand binding, and directly mediate the stabilizing or destabilizing effects on the dimerization by agonists or antagonists, respectively[37]. However, we noticed that side chains of Y281 and M252 locate right in the middle of above two proposed routes, making it unfavorable for ligands to flip them out during entry (Fig. 5b). On the other hand, within the PAS-B domain these two residues locate on the opposite side of the position corresponding to the OEA entrance in HIF-3α (between the Aβ and Cα helix as shown in Fig. 4c). This spatial distribution implies that if HIF-2α ligands enter the PAS-B pocket through a similar route as OEA into HIF-3α, it would be much easier for them to allosterically modulate the dimerization by flipping key interface residues out of the pocket (i.e., Y281 and M252). To test this hypothesis of HIF-2α possessing a similar ligand entry route, we examined whether the loop-Aβ region of HIF-2α could also switch its conformations as the corresponding region of HIF-3α by MD stimulations. RMSF values were calculated for the HIF-2α monomeric system (Fig. 5c). As shown in Fig. 5d, the loop-Aβ region of apo HIF-2α also moves outwards during the simulation, leading to a bigger and deeper pocket at the corresponding entrance position of HIF-2α PAS-B domain (Supplementary Fig. 6e, f). These MD results and above data indicated that HIF-2α may share a similar mechanism as HIF-3α,

for the entry of ligands into their PAS-B pockets and consequent allosteric effects on the dimerization.

## Discussion

The bHLH-PAS family proteins are versatile transcription factors that help living organisms to sense and respond to environmental changes[1], including the oxygen supply, day/night cycles, toxicants and so on. The physiological functions of various members of this family have been extensively studied in the past decades. For example, the oxygen-sensitive post-translational modification of HIF-α proteins by the hydroxylases, was proven as the key regulatory step for cells to adapt to different oxygen levels[4]. Meanwhile, whether any endogenous ligands can selectively bind to the HIF-α proteins and modulate their activities, has not been clear. Since the identification of a water-bound cavity within the crystal structure of HIF-2α PAS-B domain[35,40], artificial small-molecule inhibitors targeting this pocket have been discovered and developed into an FDA-approved anti-cancer drug belzutifan[41] (also known as MK-6482 and PT2977), highlighting the regulatory potential of HIF-2α by ligands. Recently we conducted a high-throughput compound screening on HIF-2α, and discovered novel HIF-2α agonists[37]. These findings suggest that potential endogenous HIF-α ligands may act as agonists or antagonists, depending on their structures and physiological roles.

Here with a screening strategy utilizing three HIF-α-ARNT heterodimer proteins, we identified an endogenous metabolite, OEA as a ligand binding selectively to the PAS-B domain of HIF-3α. OEA is a derivative of oleic acid with known effects on food intake control and lipid metabolism. Notably, the OEA concentration was reported to be around 200–400 nM in the rat small-intestine mucosa[33]. As OEA production by enterocytes elevates after feeding[28], the transient intracellular concentration of OEA may reach a micromole range. Correlating with the intestinal mobilization of OEA, HIF-3α is also highly expressed in the small intestine[42], especially in the enterocytes experiencing hypoxia due to the coverage of intestinal mucus. Interestingly, an epigenome-wide study revealed that increased DNA methylation at the HIF3A locus was associated with elevated body-mass index[43], suggesting a link between HIF-3α and body weight. Moreover, a recent genome-wide association study provided evidences that HIF-3α may function as a key regulator of lipolysis by modulating the expression levels of several related genes (e.g., LIPE, PLIN1, and PNPLA2) in the adipocytes[44], where OEA is also biosynthesized[45]. In addition, it was reported that OEA-containing supernatants from the cultured chronic lymphocytic leukemia cells induced lipolysis in isolated adipocytes[46]. These findings all imply the physiological importance of OEA as a strong candidate for endogenous HIF-3α ligands. More investigations are warranted to reveal exactly how the effects of OEA on obesity and lipolysis derive from HIF-3α relative to peroxisome proliferator-activated receptor alpha, a nuclear receptor which can be bound and activated by OEA to induce satiety[47]. The other proteins reported to interact with OEA are transient receptor potential vanilloid 1[48] and G protein-coupled receptor 119 (GPR119)[49]. Therefore, OEA may link multiple signaling pathways together and coordinate various comprehensive physiological processes besides metabolism.

Our discovery of OEA as an endogenous ligand for HIF-3α further supports the possible existence of endogenous ligand-dependent regulatory mechanisms for the HIF pathways, adding a new aspect of transcriptional modulation on top of the well-recognized oxygen-dependent hydroxylation of HIF-α subunits. Moreover, given the variation in residues outlining the PAS-B pockets of three HIF-α isoforms (Supplementary Fig. 7), each isoform is expected to recognize different endogenous ligands

selectively to regulate its activity. Furthermore, we also compared OEA with several related N-acylethanolamines (NAEs) in term of their interactions with HIF-3α, using the thermal shift assay (Supplementary Fig. 8). The results suggested that HIF-3α may not recognize palmitoylethanolamide (PEA, 16:0), arachidonoyl-ethanolamine (AEA, 20:4) or docoshexaenoylethanolamine (DHEA, 22:6), and among the 18-carbon NAEs, HIF-3α prefers OEA (18:1) to stearoylethanolamide (SEA, 18:0) and linoleoyl-ethanolamine (LEA, 18:2). This ligand specificity might be a key supplement to the more general oxygen-dependent mechanism that broadly controls the enzymatic activities of several hydroxylases, which can efficiently catalyze HIF-α proteins but with a relatively low selectivity. Therefore, these above two mechanisms working in concert would provide cells with a fine-tuned responsiveness to both oxygen level and metabolic status, emphasizing the master regulator role of HIF pathway in many biological processes.

As mentioned above, the previously reported single PAS-B domain structure of human HIF-3α came from E. coli-produced proteins, which captured 11Z-octadecenoic acid (i.e., cis-vaccenic acid; 18:1, cis-11) derivatives in the hydrophobic cavity[27]. With efforts on the removal of bacterial lipids from the purified proteins, this study also conducted a fluorescence-based binding assay that implied a preference of the HIF-3α PAS-B domain for unsaturated 18-carbon fatty acids. These findings laid the groundwork for our ongoing efforts pursuing the physiological function of HIF-3α as a specific lipid sensor, and also implied a potential mediator role for HIF-3α in connecting gut microbes and the human host. In this work, we obtained multi-domain heterodimeric HIF-3α-ARNT structures in both apo and OEA-bound forms, suggesting that lipids may not serve as mandatory structural cofactors for HIF-3α as originally proposed[27], but rather as free regulatory ligands. By directly comparing these ligand-bound HIF-3α structures (Supplementary Fig. 9a–c), we found two PAS-B domains possessing almost identical overall structures (Cα RMSD of 0.89 Å). Moreover, OEA and 1-(11Z-octadecenoyl)-sn-glycerol showed a similar shape and location within the pocket, although their carbon-carbon double bonds are indeed at different positions (cis-9 vs. cis-11) for these two monounsaturated (18:1) lipids, suggesting an accommodation of the HIF-3α PAS-B domain for various specific ligands.

Several previous studies revealed that the longer HIF-3α variants can promote tumor growth or metastasis in colorectal and pancreatic cancers[50,51], suggesting HIF-3α inhibitors might be beneficial for certain tumors. However, it has been shown that different HIF-α isoforms can either promote or suppress the tumor progression in different cancer types, such as the well-studied sibling rivalry between HIF-1α and HIF-2α isoforms[3]. Since the physiological functions of HIF-3α (especially its complicated relationship with the other isoforms) have not been fully revealed, more investigations are urgently needed to evaluate the strategy of targeting HIF-3α in tumor and other diseases. The discovery of small-molecule ligands would facilitate these studies by providing tool compounds. Therefore, we will continue to look for new HIF-3α ligands with a better selectivity and potency.

Since all the three HIF-α isoforms dimerize with ARNT in a very similar way, it is not very surprising to find that their ligands may share a common allosteric mechanism at the dimer interface. For example, here we show that for both HIF-3α and HIF-2α, agonists can stabilize the Gβ and Fα of the PAS-B domain to enhance their interactions with the ARNT A/B loop. Moreover, the ligand entry route revealed by OEA in HIF-3α might also exist in other isoforms. If so, it would provide some informative clues to design proper compounds to efficiently enter and enlarge the relatively small cavities of HIF-α, such as the one within HIF-1α PAS-B domain. Secondly, pocket entry route is one of the key

factors to consider in the design of proteolysis-targeting chimera (PROTAC) molecules[52]. Our findings may shed light on the future development of PROTAC drugs targeting HIF-α proteins.

## Methods

**Plasmid construction and site-directed mutagenesis.** For the protein over-expression in *E. coli*, mouse ARNT (GenBank AAH12870.1, residues 82–464) was cloned into the pMKH vector with or without a GFP-tag at its C-terminus as previously described[37]. Meanwhile, mouse HIF-3α (GenBank AAI20588.1, residues 4-358) was cloned into the pSJ2 vector. For the direct binding assay, the PAS-B domain of HIF-3α (residues 235-363) fused by a maltose-binding protein (MBP-HIF-3α-PAS-B) was cloned into the pSJ2 vector. For the cell-based experiments, full-length human HIF-3α1 (GenBank BAB69689.1, residues 1-667) and its mutants G237W and P490A were cloned into the pCMV-Tag4 vector. Site-directed mutagenesis was performed as instructed by the kit manufacturer, and confirmed by DNA sequencing.

**Protein expression and purification.** To obtain HIF-3α-ARNT complex proteins, the recombinant plasmids pSJ2-HIF-3α was co-transformed along with pMKH-ARNT into BL21-CodonPlus (DE3)-RIL competent cells (Agilent Technologies). The bacteria were first cultured in Luria–Bertani (LB) medium with 100 μg/ml ampicillin, 100 μg/ml kanamycin and 35 μg/ml chloramphenicol at 37 °C, and then induced overnight at 16 °C with 1 mM IPTG. Cell pellets were lysed by sonication, and supernatants were applied onto pre-packed His·Bind resin (Bestchrom). The bound proteins were further purified using SP Sepharose (GE Healthcare), and the eluted fractions were then loaded on a HiLoad 16/60 Superdex 200 pg gel-filtration column (GE Healthcare) equilibrated in buffer containing 20 mM Tris (pH 8.0) and 400 mM NaCl. DTT was added to the pooled protein peak fractions at 5 mM. The HIF-1α-ARNT and HIF-2α-ARNT complex proteins were expressed and purified as described previously[17]. The heterodimeric proteins of HIF-3α-ARNT-GFP and HIF-2α-ARNT-GFP were prepared similarly as described above, except that the pMKH-ARNT-GFP plasmid was used in the place of pMKH-ARNT. The MBP-HIF-3α-PAS-B was produced by transformation of pSJ2-MBP-HIF-3α (235-363) into BL21-CodonPlus (DE3)-RIL, followed by overnight expression and purification using His-tag affinity chromatography, as well as gel-filtration chromatography equilibrated with buffer of 20 mM HEPES (pH 7.5), 400 mM NaCl.

**Crystallization and X-ray data collection.** HIF-3α-ARNT protein complexes were crystallized by mixing equal volume of protein (4 mg/ml) and the reservoir containing sodium citrate (pH 5.5), 9% 2-propanol and 10% PEG4000 using the sitting-drop vapor diffusion method at 16 °C. Co-crystallization of HIF-3α-ARNT with OEA (100 μM) was successful in the same condition. 25% ethylene glycol was added into the reservoir solution to cryoprotect the crystals before flash frozen. Diffraction data were collected at 100 K on beamlines BL19U1 at the Shanghai Synchrotron Radiation Facility[53], and processed using the HKL3000 program[54].

**Structure determination and refinement.** The structure of HIF-3α-ARNT was solved by molecular replacement with the program Phaser[55], using the HIF-2α-ARNT structure (PDB code: 4ZP4) as the search model[17]. Further manual model building was facilitated by using Coot[56], combined with the structure refinement using Phenix[57]. Likewise, the HIF-3α-ARNT complex with OEA was solved by molecular replacement using HIF-3α-ARNT as the search model and further refined. Data collection and structure refinement statistics are summarized in Supplementary Table 1. The Ramachandran statistics, as calculated by Molprobity[58], are 98.2%/0, 96.5%/0.37% (favored/outliers) for structures of HIF-3α-ARNT and HIF-3α-ARNT-OEA, respectively. All the structural figures were prepared using PyMol (Schrödinger, LLC).

**Affinity selection-mass spectrometry (AS-MS) screening.** With both HIF-1α-ARNT and HIF-2α-ARNT proteins as counter-screening targets, we conducted the screening for HIF-3α-ARNT against an in-house compound library containing more than 2000 human endogenous metabolites, in a very similar way as previously described[37]. Briefly, the HIF-3α-ARNT protein complex was dispensed into a 96-well plate with a final concentration of 10 μM and the compound was 2 μM each. The buffer solution for screening was 20 mM Tris, 400 mM NaCl (pH 8.0) and ~1.5% DMSO. The plate was kept in a shaker at room temperature for 1 h and mixed gently. After spinning the plate to precipitate the insoluble components, the supernatant was passed through a size exclusion column to separate the protein-compound complex from the unbound compounds. The compounds bound to proteins were then heated for separation and analyzed by the TOF-MS system (Agilent Technologies).

**Thermal-shift binding assay.** To get quantitative thermal shift ($\Delta T_m$) values, we ran this assay in 96-well format using the Protein Thermal Shift (PTS) Dye Kit (Thermo Fisher Scientific 4461146) on a Quant Studio 3 qPCR machine (Thermo Fisher Scientific), according to the manufacturer's instructions. For each well, the concentration of protein complex was about 2 μM, and the compounds were tested at 20 μM in the assay buffer containing 20 mM Tris (pH 8.0), 400 mM NaCl and 2% DMSO. Melting curve data were analyzed by PTS Software V1.30 (Thermo Fisher Scientific), and $\Delta T_m$ (Derivative equation) values of different compounds were calculated accordingly.

**Surface plasmon resonance (SPR) binding assay.** To assess the binding affinity between the HIF-3α and OEA (Sigma-Aldrich O0383), the SPR assay was performed on a Biacore T200 (GE Healthcare) using commercially available CM5 sensor chips at 25 °C. The running buffer contained 10 mM HEPES (pH 7.4), 400 mM NaCl, 0.02% P-20 and 5% DMSO. According to the manufacturer's instructions, MBP and MBP-HIF-3α-PAS-B proteins diluted in 10 mM NaAc (pH 5.5) were immobilized onto the surface of flow cells 1 and 2 of the chip, respectively. Subsequently, eight different concentrations (30 μM, 25 μM, 20 μM, 10 μM, 5 μM, 2.5 μM, 1.25 μM, 0.625 μM) of OEA were prepared and injected over the chip surface with a flow rate of 30 μL/min. The time of response value was 23 s after injection start. SPR data were analyzed by BIA evaluation 3.0.2 software. The equilibrium dissociation constant $K_D$ was calculated by steady state analysis using a 1:1 Langmuir binding model.

**Time-resolved fluorescence energy transfer (TR-FRET)-based in vitro binding assay.** Protein complexes of GFP-tagged ARNT and His-tagged HIF-α were dispensed into 384-well plates with OEA of serial concentrations (0.01–200 μM). After addition of the Mab Anti-6HIS-Tb cryptate Gold (Cisbio 61H12TLF) into each well at 1.05 ng, the plate was centrifuged at $80 \times g$ for 1 min and kept in dark for 1 h. Then the protein interactions were monitored via the energy transfer signals with the Spark microplate reader (Tecan). The concentration of OEA in the control group was 0 μM, and other experimental conditions were as same as the experimental group with OEA. The TR-FRET value was determined as a ratio of the signal measured at 520 nm (GFP) to the signal measured at 492 nm (terbium). The data were analyzed in GraphPad Prism 7 as described previously[37]. The protein concentration was kept at 50 nM in the assay buffer containing 20 mM Tris (pH 7.4), 400 mM NaCl, 1 mM DTT, 0.5% Tween-20 and 2% DMSO.

**Real-time PCR.** HEK293 cells (Procell CL-0001) were cultured in DMEM medium with 10% FBS in 12-well plates at 37 °C in 5% $CO_2$. To verify the effects of HIF-3α overexpression on *HSPA6*, 0.4 μg pCMV-Tag4 or 0.4 μg pCMV-Tag4-HIF-3α1 plasmids were transfected into HEK293 cells using the jetPRIME reagent (Poly-plus-transfection). After 4 h of transfection, the medium was refreshed and the cells were cultured for another 24 h. Then the cells were harvested, and RNA was isolated using RNAiso Plus kits (TaKaRa), followed by cDNA synthesis using PrimeScript RT reagent kits (TaKaRa). To test the effects of OEA, HEK293 cells were transfected as described above and then medium was refreshed with 25 μM OEA in 0.1% DMSO. Similar procedures were taken in the case of Hep3B (Beijing Dingguo CS0172) and HepG2 (Procell CL-0103) cells, except that 0.5 μg plasmids were transfected in each well of cells. Real-time PCR was performed on a Light-Cycler 480 system (Roche) using the SYBR Green Master Mix (Yeasen). The expression of *HSPA6*, *TCF20*, *PSME2* and *IFIT1* were normalized to the expression of β-actin (*ACTB*) in the same sample. PCR primers were synthesized by PsnGene as follows: *ACTB*: (F: 5′-GCACAGAGCCTCGCCTT-3′, R: 5′-GTTGTCGACGA CGAGCG-3′); *HSPA6*: (F: 5′-GAGGTGGAGAGGATGGTTCA-3′, R: 5′-TGTC CTCTTCGGGAATCTTG-3′); *TCF20*: (F: 5′-GGTCGGTTTCAGGAATTTCA-3′, R: 5′-GCCCGCTCATAGTACTCCAG-3′); *PSME2*: (F: 5′-CCACCCAAGGATG ATGAGATG-3′, R: 5′- CAGGGACAGGACTTTCTCATTC-3′); *IFIT1*: (F: 5′-GC CCAGACTTACCTGGACAA-3′, R: 5′- GGTTTTCAGGGTCCACTTCA-3′).

**Western blotting.** HEK293 cells were cultured in DMEM medium with 10% FBS in 6-well plates at 37 °C in 5% $CO_2$, then 1 μg pCMV-Tag4, 1 μg pCMV-Tag4-HIF-3α1, or 1 μg pCMV-Tag4-HIF-3α1 (P490A) plasmids were transfected into HEK293 cells using the jetPRIME reagent when the cell density reached 70–80%. After 4 h of transfection, the medium was refreshed with or without OEA (25 μM). Another 24 h later, cells were harvested and lysed in 150 μl lysis buffer (1× TBS with 1 mM EDTA, 1% TRITON X-100 and 1× protease inhibitor cocktail) before quick spinning. After the protein concentration measurement, 40 μg of each sample was saved as input for western blotting. The primary antibodies used for detecting HIF-3α and β-actin were HIF3A Polyclonal Antibody (Proteintech, 27650-1-AP) and Beta Actin Monoclonal Antibody (Proteintech, 66009-1-Ig) respectively; and the secondary antibodies were HRP-conjugated Goat Anti-Rabbit IgG (Sangon Biotech, D110058) and HRP-conjugated Goat Anti-Mouse IgG (Sangon Biotech, D110087) respectively. The primary antibodies were used as 1:1000 dilution and the secondary antibodies were used as 1:6000 dilution.

**Molecular dynamics (MD) simulations.** The structure of OEA-bound HIF-3α-ARNT has five unresolved segments in each subunits, which were added by AutoModel class of Modeller 9.24 software[59]. The generated model with lowest value of DOPE score was further refined using Rosetta all-atom loop modelling method with the Next Generation Kinematic closure procedure, a variant of the Kinematic Closure approach. A set of 1000 loop models was generated and then clustered by backbone structural similarity. The best conformation for each loop was selected from the most populated cluster base on Rosetta energy score[60–62].

The HIF-3[noOEA] system was constructed by removing OEA from *holo* HIF-3α-ARNT with modelled segments. For the structure of *apo* HIF-3α-ARNT, the bHLH-PASA linker, FG loop, GH loop of ARNT and FG loop, HI loop of HIF-3α were obtained from modelled segments of HIF-3[OEA], the other unsolved parts were predicted as described above. For HIF-2α system, the initial structure of HIF-2α was obtained from the apo HIF-2α-ARNT complex (PDB ID: 4ZP4)[17], the unsolved segments were also predicted by Modeller and refined by Rosetta. Then, the generated structures were refined by the Protein Preparation Wizard Workflow integrated in Maestro (Version 9.0; Schrödinger, LLC) and all the parameters were set as default. The protonation states of all titratable residues and the ligand were determined PROPKA[63] at pH 7.0.

The MD simulations were carried out using Gromacs 2019.6 program package[64] with Amber ff99sb*-ILDNP force-field[65]. OEA in HIF-3[OEA] was parameterized using general Amber force-field[66] by ACPYPE tool[67]. The restricted electrostatic potential method was used to assign atomic charges of OEA at HF/6-31 G* after ab-initio optimization of the molecule. Periodic boundary conditions were applied to avoid edge effects in all calculations. Each system was solvated in a cubic box with TIP3P water molecules to keep the boundary of the box at least 10 Å away from the protein, and then neutralized with Cl ions. Minimization and equilibration were applied to all simulations. First, each system was subjected to 1000 step of steepest descent energy minimization. Then, the minimized systems were heated from 0 to 300 K by NVT MD simulations. Subsequently, a multistage NPT equilibration protocol was applied by gradually decreasing the positional restraints on all resolved atoms, backbone of resolved regions and Cα atoms. Finally, all the restraints were removed for the production runs at 300 K. The MD trajectories were analyzed by Gromacs analysis tools. The pockets during MD simulations were detected by MDpocket[68], all the parameters were set as default.

Per-residue decomposition studies of relative binding energy were carried out based on MD simulation of HIF-3[OEA] and HIF-3[noOEA] using MM-GBSA (molecular mechanics-generalized Born surface area) method, implemented in the AMBER software package[69]. In this work, the $\Delta G_{binding}$ was calculated by omitting the entropic term and therefore it is referred to as relative dimerization energy. For each system, on the basis of the equilibrated dynamic trajectory, a total of 100 snapshots were extracted from the last 10-ns trajectory with an interval of 100 ps to calculate relative dimerization energy. The binding energy of residues located at the interface between ARNT A/B-loop and HIF-3α PAS-B domain was decomposed to investigate the partial energy contributions.

**Hydrogen-deuterium exchange mass spectrometry (HDX-MS) assay**. Protein samples were processed automatically by a LEAP Technologies Hydrogen Deuterium Exchange PAL system (Carrboro). Specifically, 3.0 μL of each sample was automatically dispensed into a vial and diluted ninefold with 20 mM HEPES, 250 mM NaCl in 99.8% $D_2O$ ($pH_{read}$ 7.4) to start the deuterium exchange reaction. HDX measurements were taken at 0 s, 30 s, 100 s, 300 s, 1000 s, 3000 s and 10,000 s at 4 °C. After each time point, an aliquot of sample was transferred to a vial in a 0.5 °C chamber and quenched by addition of an equal volume of quench buffer (200 mM Citric acid, 4 M Gu-HCl, 500 mM TCEP in $H_2O$, $pH_{read}$ 2.3) for 0.5 min prior to online digestion. The complete HDX-MS procedure was repeated three times for each sample and each time point.

Each quenched sample was immediately injected into a Protease type XIII pepsin column (NovaBioAssays LLC) for 4 min at a flow rate of 50 μL/min with 0.1% formic acid in water then delivered by the loading pump on a Thermo Dionex Ultimate 3000 NCS-3500RS system (Sunnyvale). The digested peptides were trapped and desalted using a 2.1 × 5 mm Acclaim PepMap 300 C18 μ-precolumn (300 Å, 5 μm). The precolumn was connected to a 1.0 × 50 mm Thermo Hypersil Gold column C18 (175 Å, 1.9 μm). Peptides were eluted and separated by a linear gradient of Buffer B (0.1% formic acid in 80% acetonitrile) at a flow rate of 45 μL/min using the nanopump of the NCS-3500RS system. Specifically, the gradient was 4–10% over 3 min, 10–30% over 8 min, 30–90% over 1 min followed by isocratic flow with 90% Buffer B for 1 min. The online digestion, trapping, desalting process was performed at 4 °C and separation process was performed at 0.5 °C in the temperature-controlled compartment of the HDX PAL system. Data were acquired using a Thermo LTQ Orbitrap-Elite mass spectrometer with a Thermo H-ESI II probe. For peptide identification, mass spectra were acquired in a data-dependent scan using FTMS mode in MS1 (one microscan, 100 ms max injection time, 60 k resolution at 400 m/z) at the m/z range of 300–1500 followed by ten CID MS2 scans in the ion trap with a ± 2.0 m/z isolation width. Once the peptides were identified, the deuterium uptake in HDX experiments was conducted using FTMS mode in MS1.

The spectra generated were searched in PEAKS Studio X against a homemade database including target protein with a precursor mass tolerance of ≤20 ppm and MS/MS fragment ≤0.02 Da. Retention time and sequence information for each peptide were exported to Excel for HDX data processing.

HDX data analysis was carried out using HDExaminer 2.0 (Sierra Analytics Inc.). The number of D taken up (D-uptake) by each peptide at each exchange time was calculated by the software algorithm for matching the best theoretical isotope distribution pattern to the observed isotope distribution pattern. D-uptake was plotted as a function of exchange time. Triplicate runs were compared using Student's *t* test at the 95% confidence level to confirm the consistency of the analytical results obtained. D-uptake was converted to %D for each peptide based on the theoretical number of D; %D was used to generate heat maps, butterfly

comparisons, and difference plots. In addition, HDX-MS analysis was also carried out on non-deuterated and fully deuterated samples to correct back-exchange[70].

**Statistical analysis**. All statistical data were calculated using GraphPad Prism version 7.0. The sequence alignment figure of three HIF-α proteins was generated by ESPript 3.0. The pocket volume calculation was conducted using Fpocket 4.0 and PyVOL 1.7.6. Unpaired two-tailed *t*-test was used to compare the means of two groups. Significance of mean comparison is annotated as follow: *$p < 0.05$; **$p < 0.01$; ***$p < 0.001$, and $p$ value of <0.05 was considered to be statistically significant.

**Reporting summary**. Further information on research design is available in the Nature Research Reporting Summary linked to this article.

## Data availability

The data that support the findings of this study are available from the corresponding authors upon reasonable request. Coordinates and structure factors of the HIF-3α-ARNT protein complexes in apo and OEA-bound forms, have been deposited in the Protein Data Bank under accession codes 7V7L and 7V7W, respectively. The accession codes for the previously released structures HIF-2α-ARNT, HIF-2α-ARNT-PT2385, HIF-2α-ARNT-M1001 and HIF-3α PAS-B-1-(11Z-octadecenoyl)-sn-glycerol are 4ZP4, 6E3S, 6E3U and 4WN5, respectively. The HDX-MS data are available under accession code PXD033376 in the PRIDE database. Source data are provided with this paper.

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

## Acknowledgements

We thank Zhifeng Li, Jing Zhu and Xiaoju Li from State Key laboratory of Microbial Technology of Shandong University for their help and guidance in the experiments of SPR, Real-time PCR and X-ray diffraction. We also thank the staffs from BL19U1 beamline of National Facility for Protein Science in Shanghai (NFPS) at Shanghai Synchrotron Radiation Facility, for their assistance during data collection. This work was supported by grants from National Natural Science Foundation of China (22177063), Shandong Provincial Natural Science Foundation, China (ZR2021JQ30), National Key R&D Program of China (2018YFE0113000), Taishan Scholars Program of Shandong Province (tsqn201909004), and State Key Laboratory of Drug Research (SIMM2105KF-03) to D.W., grants from National Natural Science Foundation of China (81625022, 91853205 and 81821005), National Key R&D Program of China (2021ZD0203900), Science and Technology Commission of Shanghai Municipality (19XD1404700), and the project of National Multidisciplinary Innovation Team of Traditional Chinese Medicine

supported by National Administration of Traditional Chinese Medicine to C.L., a grant from National Institutes of Health (R01DK118297) to F.R., and grants from Science and Technology Commission of Shanghai Municipality (18DZ2210200), Construction and Operation of Zhangjiang Laboratory (II) (19DZ2260100) to C.P.

## Author contributions

X.D. conducted experiments including clone construction, protein purification, crystallization, X-ray diffraction, structure refinement and binding assays; F.Y. performed the MD simulations; M.Z. and J.Z. conducted the cell-based experiments; X.R. and X.C. participated in the protein purification and crystallization; X.T. and C.P. executed the HDX-MS analysis; J.L. handled the compound screening; X.S. participated in the plasmid construction; Z.H. and H.D. assisted in the binding assays; F.L. participated in the diffraction data collection and structure refinement; D.W., C.L., and F.R. conceived the study, supervised experiments, and wrote the paper with inputs from all authors.

## Competing interests

F.R. is a founder and consultant for Flare Therapeutics. The remaining authors declare no competing interests.
