## [Peer Review File · Nature Communications]

Identification of oleoylethanolamide as an endogenous ligand for HIF-3 αReviewers' Comments:

Reviewer #1:

Remarks to the Author:

Diao et al. provide a combination of structural biology, biochemistry, and cell biology studies of the HIF-3a:ARNT (=HIF-3) heterodimer, which is a relatively poorly-characterized member of the bHLH-PAS transcription factor family. Starting from a crystal structure of an "apo" form of HIF-3 showing a 280 Å³ cavity in the middle of the HIF-3a PAS-B domain, analogous to prior findings in HIF-2a, the authors proceed to screen a metabolite library to identify OEA as a natural (and markedly specific) ligand for this cavity which can markedly improve the strength and functional potency of the HIF-3 heterodimer. The mechanism of this stabilization is studied with a combination of X-ray crystallography, HDX-MS, and MD, with a reasonable model presented involving a key linker between the ARNT PAS-A and -B domains.

Overall, this manuscript provides a nice advance to this area of ligand-regulated transcription factors, giving a number of interesting insights in an area attracting attention for both artificial and natural control of gene expression. These strengths are currently offset by several issues that should be addressed in revision:

Substantial changes:

- I am very surprised by the authors' lack of comparison between their work and the prior HIF-3a PAS-B structure (and biochemical analyses) reported by Fala et al in 2015, ref. 28. Despite differences in experimental route and conditions, that paper identified 1). Binding of 18:1 fatty acids and derivatives by HIF-3a PAS-B (with some accompanying SAR information on lipid), 2). The structure of a HIF-3a PAS-B:lipid complex showing binding of lipid into the internal cavity, and 3). Stabilization of HIF-3a PAS-B:ARNT PAS-B interactions by lipid. These are three substantial points that are not properly acknowledged in this work as currently written. To be clear, while I think Diao et al. has some made several substantial advantages over the prior report, specifically in their use of larger bHLH-PAS-PAS constructs, obtaining an important lipid-free structure, and obtaining cellular data of activation, I think the current manuscript makes a major omission by not more fully acknowledging the prior work (beyond the current text on l. 101-105) and comparing their results.

- on a related front, I encourage the authors to carefully review any comments that may seem to indicate priority regarding the potential for endogenous small molecule ligands in regulating HIF activities directly. Fala et al. 2015 made a strong case for this in their paper from almost seven years ago; such commentary has been widespread in the field as well for some time given the findings in HIF-2a with a large internal cavity. The authors are entitled to their opinion on this matter, of course, and are welcome to share it – but they should at least directly comment on prior work as they do.

- I strongly encourage the authors to more regularly state ligand concentrations used in various assays at the points where these are discussed in the manuscript text and figure legends. These are often found solely in Methods, complicating more easy comparisons across different assays in different panels/figures or different spots in the text.

Minor changes:

- l. 133-134: ligand binding into HIF-2a PAS-B cavity affecting interaction w/ARNT with functional effects shown well before the 2019 ref 35 stated – ref 37 (2009) and Scheuermann et al. Nat Chem Biol (2013).

- Fig 2: absolutely essential to list OEA concentrations for 2e, which has no such information provided in text (146-152) or Methods. TR-FRET assay very confusing on how to calibrate "100%" heterodimer improvement.

- l. 165-167, 186-189: the increase in cavity size from 280 to 700 Å³ is quite dramatic, but not

entirely clear where this originates from given Fig 3b-d. Subsequent statements about minimal change in pocket-lining sidechains also confuse this issue a bit. Perhaps a comparison of sidechain positions in apo- and OEA-bound forms (e.g. Fig 3b) would help? How do these compare to the results of Fala 2015?

- I. 260-280: the authors' arguments about HIF-2a M252 and Y281 "flipping out during entry" are difficult to follow. Is it not possible that a type of conformational selection occurs with M252 and Y281 "flipping out" prior to ligand binding? Is this seen in the ref 36 or authors' own MD work?

- Ext Data Fig 5: HDX-MS data here suggest several regions of ARNT with substantial OEA-dependent effects (210-220, 305-315, 380, 410); where are these on structure, and can they be rationalized? Additionally, why no OEA effect in ARNT A/B loop?

- Figure legends are generally light on detail; in addition to ligand concentrations mentioned above, I encourage the authors to review to ensure that protein chains (and colors) are properly specified for each panel.

Reviewer #2:

Remarks to the Author:

This MS by Diao et al. reports that OEA, a cellular metabolite, can bind to the HIF3a-ARNT specifically. Based on structural and functional analysis results, they proposed that OEA binding can enhance the HIF3a-ARNT dimerization, stability, and increase its activity in regulating gene expression. The authors also provided evidence pointing to the entry route and to an allosteric mechanism that enhances the HIFa-ARNT stability. These findings are new and potentially important. Overall, the MS is well written and the experiments appeared to be performed competently. There are, however, several issues.

Specific comments

#1) One of the main conclusions is that OEA binds to the PAS-B domain of HIF3a and this increases HIF3a-ARNT dimer formation and stability. Given the screening was performed using preformed HIF3a-ARNT dimers (Fig. 1) and that OEA clearly can bind preformed HIF3a-ARNT dimers (Fig. 2B), this may not be entirely correct. Please clarify.

#2) The only functional experiment (Fig. 2e) was performed in HEK293 cells under normal O₂. Under this setting, most HIF3a would presumably be degraded given that the authors expressed a wild-type HIF3a rather than a stabilized mutant HIF3a. Is the increased gene expression reflects a tighter and stable HIF3a-ARNT dimer or due to stable and less degradation or both? A simple Western blot would be helpful to answer.

#3) A comparison of the biological/transcriptional activity of the G237W mutant with wild-type HIF3a should be performed. This will test the notion that OEA binding can enhance the HIF3a-ARNT dimerization, stability, and transcriptional activity.

#4) What is the physiological levels of OEA? Can it enter the cells? Is 14 μ M considered physiological?

Reviewer #3:

Remarks to the Author:

The presented study by Wu and colleagues identified oleoylethanolamide (OEA) as novel highly selective ligand for HIF3a-ARNT heterodimer proteins. Detailed biophysical characterization using thermal shift assay, surface plasma resonance and time-resolved fluorescence resonance energy

transfer confirmed the direct binding between OEA and the PAS-B domain pocket of HIF3a and showed that OEA binding enhanced interaction between HIF3a and ARNT. Elevated mRNA level of HSPA6 suggested increased transcriptional activity of HIF3a upon OEA binding. Subsequently, the authors solved the crystal structure of HIF3a-ARNT heterodimer in complex with OEA to understand the molecular details of how OEA increases HIF3a activity by promoting dimerization with ARNT. Based on the structure and further analyses, it was concluded that OEA allosterically enhances dimerization by stabilizing key residues within the G-bb and F-a regions of HIF3a PAS domain. Also the ligand entry route into the PAS-B pocket could be described. Finally, a common allosteric effect on dimerization of HIFa isoforms by ligand binding was suggested.

Overall, the study is of high quality and describes a very interesting, new and topical finding that is presented in clear figures; it shows that the endogenous cellular metabolite OEA directly binds HIF3a and regulates dimerization with ARNT. Many complementary biophysical approaches were used to substantiate the structural and molecular details of this interaction. However, the physiological relevance of OEA binding and its agonistic effect in cellulo is only shown by mRNA expression level of a single HIF3a target gene in a HIF3a overexpression scenario. I recommend to analyse additional target genes to strengthen this part of the manuscript before publication. Furthermore, as HIF3a is highly expressed in the small intestine (and OEA mobilized there upon feeding), the authors could use a corresponding cell line and test the effect of OEA treatment on the activity of endogenous HIF3a.

We thank all three reviewers for their careful evaluation, strong overall encouragement, and constructive guidance in improving our manuscript.

Reviewer #1 (Remarks to the Author):

Diao et al. provide a combination of structural biology, biochemistry, and cell biology studies of the HIF-3a:ARNT (=HIF-3) heterodimer, which is a relatively poorly-characterized member of the bHLH-PAS transcription factor family. Starting from a crystal structure of an “apo” form of HIF-3 showing a 280 Å³ cavity in the middle of the HIF-3a PAS-B domain, analogous to prior findings in HIF-2a, the authors proceed to screen a metabolite library to identify OEA as a natural (and markedly specific) ligand for this cavity which can markedly improve the strength and functional potency of the HIF-3 heterodimer. The mechanism of this stabilization is studied with a combination of X-ray crystallography, HDX-MS, and MD, with a reasonable model presented involving a key linker between the ARNT PAS-A and -B domains.

Overall, this manuscript provides a nice advance to this area of ligand-regulated transcription factors, giving a number of interesting insights in an area attracting attention for both artificial and natural control of gene expression. These strengths are currently offset by several issues that should be addressed in revision:

Substantial changes:

- I am very surprised by the authors' lack of comparison between their work and the prior HIF-3a PAS-B structure (and biochemical analyses) reported by Fala et al in 2015, ref. 28. Despite differences in experimental route and conditions, that paper identified 1). Binding of 18:1 fatty acids and derivatives by HIF-3a PAS-B (with some accompanying SAR information on lipid), 2). The structure of a HIF-3a PAS-B:lipid complex showing binding of lipid into the internal cavity, and 3). Stabilization of HIF-3a PAS-B:ARNT PAS-B interactions by lipid. These are three substantial points that are not properly acknowledged in this work as currently written. To be clear, while I think Diao et al. has some made several substantial advantages over the prior report, specifically in their use of larger bHLH-PAS-PAS constructs, obtaining an important lipid-free structure, and obtaining cellular data of activation, I think the current manuscript makes a major omission by not more fully acknowledging the prior work (beyond the current text on l. 101-105) and comparing their results.

Many thanks for these critical comments. Accordingly, we have modified our manuscript to acknowledge the significant findings by Fala et al in their 2015 paper. Please check the following sentences added into the new manuscript in blue color (Introduction, Page 4, Lines 81-83; Discussion, Pages 15-16, Lines 358-370). To better illustrate the common features and differences between our multi-domain OEA-bound HIF-3 α -ARNT structure and the 1-(11Z-octadecenoyl)-sn-glycerol-bound HIF-3 α PAS-B domain structure, we also added a new figure (Extended Data Fig. 9) to accompany the comparison.

- on a related front, I encourage the authors to carefully review any comments that may seem to indicate priority regarding the potential for endogenous small molecule ligands in regulating HIF activities directly. Fala et al. 2015 made a strong case for this in their paper from almost seven years ago; such

commentary has been widespread in the field as well for some time given the findings in HIF-2a with a large internal cavity. The authors are entitled to their opinion on this matter, of course, and are welcome to share it – but they should at least directly comment on prior work as they do.

As mentioned in our answer to the first question, we have modified the manuscript by adding sentences acknowledging the great findings by Fala et al (Introduction, Page 4, Lines 81-83; Discussion, Pages 15-16, Lines 358-370).

- I strongly encourage the authors to more regularly state ligand concentrations used in various assays at the points where these are discussed in the manuscript text and figure legends. These are often found solely in Methods, complicating more easy comparisons across different assays in different panels/figures or different spots in the text.

Thank you for this valuable suggestion. We have stated the concentrations of OEA in the manuscript text and figure legends.

Minor changes:

- l. 133-134: ligand binding into HIF-2a PAS-B cavity affecting interaction w/ARNT with functional effects shown well before the 2019 ref 35 stated – ref 37 (2009) and Scheuermann et al. Nat Chem Biol (2013).

This point has been rephrased in the manuscript (Pages 6-7, Lines 136-139).

- Fig 2: absolutely essential to list OEA concentrations for 2e, which has no such information provided in text (146-152) or Methods. TR-FRET assay very confusing on how to calibrate “100%” heterodimer improvement.

We have listed the concentrations of OEA in the figure legends and methods. The calculation formula of heterodimerization improvement used in the TR-FRET assay is as follows:

$$\frac{\text{Ratio of OEA groups (520 nm/492 nm)} - \text{Ratio of control group (520 nm/492 nm)}}{\text{Ratio of control group (520 nm/492 nm)}}$$

- l. 165-167, 186-189: the increase in cavity size from 280 to 700 Å³ is quite dramatic, but not entirely clear where this originates from given Fig 3b-d. Subsequent statements about minimal change in pocket-lining sidechains also confuse this issue a bit. Perhaps a comparison of sidechain positions in apo- and OEA-bound forms (e.g. Fig 3b) would help? How do these compare to the results of Fala 2015?

According to this constructive suggestion, we have reorganized related descriptions about the cavity size in the manuscript (Pages 8-9, Lines 187-196 and 211-213). We compared the conformations of key residues in the HIF-3α PAS-B pockets in both “apo” and OEA-bound forms with several new figure panels (Extended Data Fig. 3c-e).

- 1. 260-280: the authors' arguments about HIF-2 α M252 and Y281 "flipping out during entry" are difficult to follow. Is it not possible that a type of conformational selection occurs with M252 and Y281 "flipping out" prior to ligand binding? Is this seen in the ref 36 or authors' own MD work?

In our 500-ns MD stimulation of "apo" HIF-2 α protein, we checked the conformations of M252 and Y281 (as shown in the Figure R1 below) and found that side-chains of these two residues indeed show some vibrations, especially for Y281. However, their side-chains all stay within the PAS-B pocket during the whole stimulation. Therefore, we believe the "side-chain flipping out" phenomena of M252 and Y281 are caused by antagonists and agonists during their entry of pocket, respectively.

Figure R1. A 500-ns MD stimulation of HIF-2 α protein highlighting the M252 and Y281 residues.

- Ext Data Fig 5: HDX-MS data here suggest several regions of ARNT with substantial OEA-dependent effects (210-220, 305-315, 380, 410); where are these on structure, and can they be rationalized? Additionally, why no OEA effect in ARNT A/B loop?

These ARNT regions locate in the PAS-A or PAS-B domains relatively far away from the HIF-3 α residues interacting with OEA. Their H/D exchanges would be influenced by OEA binding through allosteric effects along the protein complex. OEA showed a weak effect on promoting the stability of ARNT A/B loop (347-360) at 30 s (Extended Data Fig. 5a), but not obvious at other time points in the HDX-MS. The possible reason is that loop regions tend to have a poor stability and a high exchange level of hydrogen and deuterium compared with the regions containing solid secondary structures. Therefore, it is difficult to accurately compare the H/D exchanges caused by OEA on the ARNT A/B loop. This was also found in our previous study (ref 37). The HIF-2 α agonist M1001 could stabilize the ARNT A/B loop in the crystal structure, but showed no obvious stabilizing effect in the HDX-MS data.

- Figure legends are generally light on detail; in addition to ligand concentrations mentioned above, I encourage the authors to review to ensure that protein chains (and colors) are properly specified for each panel.

Thank you for the good advice. We have described the colors for each protein/ligand, along with ligand concentrations in the figure legends.

Reviewer #2 (Remarks to the Author):

This MS by Diao et al. reports that OEA, a cellular metabolite, can bind to the HIF3a-ARNT specifically. Based on structural and functional analysis results, they proposed that OEA binding can enhance the HIF3a-ARNT dimerization, stability, and increase its activity in regulating gene expression. The authors also provided evidence pointing to the entry route and to an allosteric mechanism that enhances the HIFa-ARNT stability. These findings are new and potentially important. Overall, the MS is well written and the experiments appeared to be performed competently. There are, however, several issues.

Specific comments

#1) One of the main conclusions is that OEA binds to the PAS-B domain of HIF3a and this increases HIF3a-ARNT dimer formation and stability. Given the screening was performed using preformed HIF3a-ARNT dimers (Fig. 1) and that OEA clearly can bind preformed HIF3a-ARNT dimers (Fig. 2B), this may not be entirely correct. Please clarify.

Many thanks for the comments. We performed the compound screening using three HIF- α -ARNT heterodimers, mainly due to the reason that multi-domain HIF- α proteins could only be expressed in complex with the multi-domain ARNT. Here we used HIF-1 α -ARNT and HIF-2 α -ARNT as controls to identify ligands that specifically bind to HIF-3 α . We think that OEA can bind to either monomeric or dimeric HIF-3 α , and enhance the stability of HIF-3 α -ARNT dimer. Formation of this dimer may not rely on ligand binding, as in the case of HIF-2 α -ARNT.

#2) The only functional experiment (Fig. 2e) was performed in HEK293 cells under normal O₂. Under this setting, most HIF3a would presumably be degraded given that the authors expressed a wild-type HIF3a rather than a stabilized mutant HIF3a. Is the increased gene expression reflects a tighter and stable HIF3a-ARNT dimer or due to stable and less degradation or both? A simple Western blot would be helpful to answer.

According to this nice suggestion, we have performed the WB experiment (Extended Data Fig. 2d) and added a paragraph of descriptions in the manuscript (Pages 7-8, Lines 162-171). There is only one hydroxylation site in the oxygen-dependent degradation domain (ODDD) of HIF-3 α , possibly rendering it less sensitive to oxygen than HIF-1 α and HIF-2 α . In our experiment, a degradation-resistant mutation (P490A) was introduced for HIF-3 α . The results showed that the basal level of HIF-3 α protein was low in HEK293 cells. And the levels of HIF-3 α 1 (WT) and HIF-3 α 1 (P490A) were similar after overexpression. Moreover, OEA had little effect on the protein levels of HIF-3 α 1. These results may suggest that OEA could increase downstream gene expression by promoting dimer stability and have little effect on HIF-3 α protein degradation.

#3) A comparison of the biological/transcriptional activity of the G237W mutant with wild-type HIF3a should be performed. This will test the notion that OEA binding can enhance the HIF3a-ARNT dimerization, stability, and transcriptional activity.

Thank you very much for the good advice. We have conducted this experiment (new Fig. 4f) and modified the text accordingly (Page 11, Lines 252-253). Similar to the wild-type, HIF-3 α G237W mutant could promote the expression of *HSPA6* gene. However, OEA treatment could not further increase the expression significantly for G237W.

#4) What is the physiological levels of OEA? Can it enter the cells? Is 14 μ M considered physiological?

The concentrations of OEA were reported around 5~10 nM and 200~400 nM in the plasma and small intestine of rat, respectively (ref 33, Fu et al, J Biol Chem 2007). For human, the concentration of OEA in plasma was about 40 nM (Psychogios et al, PLoS One 2011 Feb 16; 6: e16957). However, we could not find information about its concentration in human small intestine or inside cells. As OEA produced by intestinal epithelial cells elevates significantly after feeding (ref 33), the transient concentration of OEA may reach a micromole range, close to the K_D value we measured for OEA binding to HIF-3 α . There were many studies using OEA for cell-based experiments in the past, showing the ability of OEA entering cells. For example, Fu et al. treated Hela cells with OEA and proved the agonistic activity of OEA on PPAR α (ref 47, Fu et al, Nature 2003).

Reviewer #3 (Remarks to the Author):

The presented study by Wu and colleagues identified oleoylethanolamide (OEA) as novel highly selective ligand for HIF3a-ARNT heterodimer proteins. Detailed biophysical characterization using thermal shift assay, surface plasma resonance and time-resolved fluorescence resonance energy transfer confirmed the direct binding between OEA and the PAS-B domain pocket of HIF3a and showed that OEA binding enhanced interaction between HIF3a and ARNT. Elevated mRNA level of *HSPA6* suggested increased transcriptional activity of HIF3a upon OEA binding. Subsequently, the authors solved the crystal structure of HIF3a-ARNT heterodimer in complex with OEA to understand the molecular details of how OEA increases HIF3a activity by promoting dimerization with ARNT. Based on the structure and further analyses, it was concluded that OEA allosterically enhances dimerization by stabilizing key residues within the G-bb and F-a regions of HIF3a PAS domain. Also the ligand entry route into the PAS-B pocket could be described. Finally, a common allosteric effect on dimerization of HIFa isoforms by ligand binding was suggested.

Overall, the study is of high quality and describes a very interesting, new and topical finding that is presented in clear figures; it shows that the endogenous cellular metabolite OEA directly binds HIF3a and regulates dimerization with ARNT. Many complementary biophysical approaches were used to substantiate the structural and molecular details of this interaction. However, the physiological relevance of OEA binding and its agonistic effect in cellulo is only shown by mRNA expression level of a single HIF3a target gene in a HIF3a overexpression scenario. I recommend to analyse additional target genes to strengthen this part of the manuscript before publication.

We appreciate the comments and valuable advice. As mentioned in the manuscript, we tested several previously reported downstream genes of HIF-3 α , including *EPO*, *HSPA6*, *REDD1*,

LC3C, *SQRDL* and *ANGPT4* in the HEK293 and HEP3B cells. Unfortunately, no other genes were up-regulated by overexpressing HIF-3 α 1, except *HSPA6* (Figure R2). We speculated that these results might be due to different variants of HIF-3 α or differences in experimental conditions, such as the culture medium. Therefore, following the reviewer's advice, we then conducted a preliminary RNA-seq to explore potential downstream genes by overexpressing HIF-3 α 1 in HEK293 cells. We found that in addition to *HSPA6*, overexpression of HIF-3 α 1 could promote the expression of multiple genes such as *TCF20*, *PSME2* and *IFIT1* (as shown in Extended Data Fig. 2c and Lines 157-161). And OEA further increased the mRNA levels of these genes, verifying the effect of OEA on HIF-3 α as an agonist.

Figure R2. The analysis of previously reported HIF-3 α downstream genes.

Furthermore, as HIF3 α is highly expressed in the small intestine (and OEA mobilized there upon feeding), the authors could use a corresponding cell line and test the effect of OEA treatment on the activity of endogenous HIF3 α .

Thanks for this suggestion. Due to the unavailability of ethical approvals in a short time, we were unable to conduct this experiment using human- or mouse-derived primary cells. The only related cell line we could obtain was a mouse intestinal epithelial cell line called MODE-K. We treated MODE-K cells with OEA and measured the mRNA expression of potential HIF-3 α downstream genes by PCR, including *EPO*, *HSPA1b* (a mouse homolog gene of *HSPA6*), *LC3C*, *REDD1*, *SQRDL*, *ANGPT4*, *TCF20*, *PMSE2* and *IFIT1*. The mRNA levels of *EPO* and *LC3C* were too low to be detected; and OEA showed no significant effect on the other genes, except for *IFIT1* whose expression was surprisingly reduced by OEA (Figure R3). In our next work, we plan to further study the signaling pathways regulated by OEA through HIF-3 α in more types of small intestinal epithelial cells, as well as in the HIF-3 α KO mice.

Figure R3. The effects of OEA (25 μ M) on HIF-3 α downstream genes in normoxia.

Reviewers' Comments:

Reviewer #1:

Remarks to the Author:

Diao et al. have markedly improved this manuscript with changes in response to my inquiries and those of the other reviewers. I suggest only three minor points before publication:

- 1). R1 comment re: comparison to Fala et al. 2015 (Discussion, pages 15-16, lines 358-370) – nice addition to the manuscript, but I think the insight provided by Fala et al. 2015 is still not quite fully stated here. I simply encourage the authors to acknowledge that this prior work investigated the binding of a variety of different lipids to HIF-3a PAS-B with a variety of biochemical methods, finding diversity in both the location of unsaturation in the acyl chain (including high affinity binding of a 18:2 linoleic acid!) and both neutral and phospholipids bound. Some ability to exchange lipids was also reported, with impact on protein binding affinity.
- 2). R1 comment re: comparison to Fala et al. 2015 (Extended Data Fig 9) – also nice addition to the manuscript, but should cite Fala et al 2015 (and provide PDB code) somewhere here to unambiguously identify where the PAS-B structure originated from.
- 3). R2 comment re: in vivo concentration of OEA – the reviewer's question and authors' response were intriguing. I think this would be of broad interest to the readers of this work, maybe consider putting some brief summary of this in the Discussion?

Reviewer #2:

Remarks to the Author:

This MS by Diao et al. reports that OEA, a cellular metabolite, can bind to the HIF3a-ARNT specifically. Based on structural and functional analysis results, they proposed that OEA binding can enhance the HIF3a-ARNT dimerization, stability, and increase its activity in regulating gene expression. The authors also provided evidence pointing to the entry route and to an allosteric mechanism that enhances the HIFa-ARNT stability. These findings are new and potentially important.

The revised MS has addressed my major comments.

Reviewer #3:

Remarks to the Author:

All comments and concerns raised by the reviewers have been sufficiently addressed by the authors. I recommend publication without delay.

Again, we thank all the reviewers and editors for the constructive suggestions and guidance we received to improve this paper.

Reviewer #1 (Remarks to the Author):

Diao et al. have markedly improved this manuscript with changes in response to my inquiries and those of the other reviewers. I suggest only three minor points before publication:

1). R1 comment re: comparison to Fala et al. 2015 (Discussion, pages 15-16, lines 358-370) – nice addition to the manuscript, but I think the insight provided by Fala et al. 2015 is still not quite fully stated here. I simply encourage the authors to acknowledge that this prior work investigated the binding of a variety of different lipids to HIF-3a PAS-B with a variety of biochemical methods, finding diversity in both the location of unsaturation in the acyl chain (including high affinity binding of a 18:2 linoleic acid!) and both neutral and phospholipids bound. Some ability to exchange lipids was also reported, with impact on protein binding affinity.

As suggested, we added one more sentence in the Discussion section to acknowledge the binding assay by Fala et al. 2015 (Page 16, Lines 363-365).

2). R1 comment re: comparison to Fala et al. 2015 (Extended Data Fig 9) – also nice addition to the manuscript, but should cite Fala et al 2015 (and provide PDB code) somewhere here to unambiguously identify where the PAS-B structure originated from.

The reference and PDB code have been added into the legends of this figure.

3). R2 comment re: in vivo concentration of OEA – the reviewer’s question and authors’ response were intriguing. I think this would be of broad interest to the readers of this work, maybe consider putting some brief summary of this in the Discussion?

According to this advice, we briefly discussed the physiological concentration of OEA in the Discussion section (Page 14, Lines 326-329).

Reviewer #2 (Remarks to the Author):

This MS by Diao et al. reports that OEA, a cellular metabolite, can bind to the HIF3a-ARNT specifically. Based on structural and functional analysis results, they proposed that OEA binding can enhance the HIF3a-ARNT dimerization, stability, and increase its activity in regulating gene expression. The authors also provided evidence pointing to the entry route and to an allosteric mechanism that enhances the HIFa-ARNT stability. These findings are new and potentially important.

The revised MS has addressed my major comments.

Many thanks for the comments.

Reviewer #3 (Remarks to the Author):

All comments and concerns raised by the reviewers have been sufficiently addressed by the authors. I recommend publication without delay.

Many thanks for the comments.